# Mechanism of Histone Arginine Methylation Dynamic Change in Cellular Stress

**DOI:** 10.3390/ijms25147562

**Published:** 2024-07-10

**Authors:** Xiao-Guang Ren, Wei Li, Wen-Xuan Li, Wen-Qiang Yu

**Affiliations:** Department of RNA Epigenetics, Faculty of Institute of Biomedical Sciences, Campus of Shanghai Medical College, Fudan University, Shanghai 200032, China; 16111510049@fudan.edu.cn (X.-G.R.); liwei_epi@fudan.edu.cn (W.L.); 20111510022@fudan.edu.cn (W.-X.L.)

**Keywords:** H3R26me2s, H3K27ac, stress, crosstalk

## Abstract

Histone arginine residue methylation is crucial for individual development and gene regulation. However, the dynamics of histone arginine methylation in response to cellular stress remains largely unexplored. In addition, the interplay and regulatory mechanisms between this and other histone modifications are important scientific questions that require further investigation. This study aimed to investigate the changes in histone arginine methylation in response to DNA damage. We report a global decrease in histone H3R26 symmetric dimethylation (H3R26me2s) and hypoacetylation at the H3K27 site in response to DNA damage. Notably, H3R26me2s exhibits a distribution pattern similar to that of H3K27ac across the genome, both of which are antagonistic to H3K27me3. Additionally, histone deacetylase 1 (HDAC1) may be recruited to the H3R26me2s demethylation region to mediate H3K27 deacetylation. These findings suggest crosstalk between H3R26me2s and H3K27ac in regulating gene expression.

## 1. Introduction

Methylation of histone arginine residues is a key epigenetic modification, with mutations at the H3R2, H3R8, and H3R26 sites linked to developmental lethality [1], highlighting the critical role of these arginine sites in cellular function. Unlike lysine residue modifications, arginine residues have two isoform methylation states, symmetric or asymmetric demethylation, each with distinct functions in regulating gene expression. H3R2me2a, H3R8me2s, and H4R3me2s have been reported to repress gene expression, while H3R2me2s, H3R8me2a, and H4R3me2a act as active markers [2,3,4]. At the H3R26 site, H3R26me2a was reported to be catalyzed by the protein arginine methyltransferase 4 (PRMT4), which is enriched in gene promoters and activates gene expression [5,6]. H3R26me2a was also shown to regulate mouse embryo development at an early stage [7]. However, the functions of H3R26me2s and its associated enzymes remain largely unknown.

Notably, histone residues H3R2, H3R8, H3R26, and H4R3 are proximal to H3K4, H3K9, H3K27, and H4K5. Weimin et al. reported that the H3R26A mutation in *Drosophila melanogaster* caused H3K27me3 depletion. However, this study was unable to distinguish which modifications of H3R26me2a and H3R26me2s caused H3K27me3 depletion [1]. PRMT4 catalyzes H3R26me2a, which is enriched in gene promoters and activates gene expression [5,6]. H3R26me2a blocks polycomb repressive complex 2 (PRC2) binding; however, more direct evidence is needed [8]. Another study has shown that H3K27ac stimulates H3R26me2a, whereas H3K27me3 blocks it [9]. H4K5ac regulates PRMT1 and PRMT5 for H4R3 methylation [10]. In addition, H4K5 acetylation regulates PRMT1 and PRMT5 for H4R3 methylation [11,12,13]. Methylation of H3R2me2a blocks the binding of the mixed lineage leukemia protein 1 (MLL1) complex to the H3K4 site, catalyzing H3K4 trimethylation [11]. In contrast, H3R2me2s is strongly associated with H3K4me3 [14]. These findings reveal a complex interplay between histone arginine methylation and lysine methylation or acetylation. However, cooperative mechanisms between symmetric and asymmetric histone arginine methylation and adjacent lysine modifications are not fully understood.

Histone modifications exhibit dynamic changes in response to DNA damage [15,16,17,18,19]. For example, lysine undergoes deacetylation following DNA damage, leading to reduced levels of H4K16ac and H3K56ac in cell lines treated with ultraviolet (UV) light, heat, or IR [20,21,22]. HDAC1-2 mediates deacetylation at H3K56, potentially aiding DNA repair via the non-homologous end joining (NHEJ) pathway. Conversely, H3K56 hyperacetylation is associated with spontaneous DNA damage [23,24]. H4K16ac is an important modification in the DNA damage response [19,25,26]. Decreased H4K16ac with heat treatment is mediated by recruiting sirtuin 1 (SIRT1), which impairs the homologous recombination (HR)-mediated DNA damage response [22].

In contrast to modifications of histone lysine, the distribution and dynamic changes in most histone arginine residue methylations in response to DNA damage remain largely unexplored. Although the distribution characteristics of H3R2me2s, H4R3me2a, and H4R3me2s modifications have been analyzed [27,28], the distribution of other arginine residues has not been fully investigated. Thus, it remains unclear how these modifications change dynamically in response to DNA damage.

To investigate the changes in histone arginine methylation in response to DNA damage, we induced DNA damage using bleomycin (BLM). We observed a decrease in symmetric arginine methylation but not asymmetric methylation, accompanied by a reduction in histone lysine acetylation but not methylation. Additionally, we found that H3R26me2s co-localized with H3K27ac, but H3R26me2s and H3K27me3 are mutually exclusive. Interestingly, H3R26me2s was found to modulate the state of H3K27ac, while changes in H3K27ac levels did not affect the state of H3R26me2s. This was confirmed using EIA binding protein 300 (P300), HDAC, or PRMT5 inhibitors. Moreover, we found that H3R26me2s demethylation may recruit HDAC1 and mediate H3K27ac deacetylation. Finally, H3R26me2s and H3K27ac collaborated to regulate gene expression in response to DNA damage. These results suggest a crosstalk between H3R26me2s and H3K27ac in the DNA damage response.

## 2. Results

### 2.1. Results (Main Text)

#### 2.1.1. Symmetric Methylation of H3R26 Specifically Responds to Stress-Induced DNA Damage

First, we investigated the dynamic changes in histone arginine methylation following DNA damage in cells. We induced DNA damage in cells using BLM and observed an increase in the DNA damage marker γH2A.X in both HepG2 and LM3 cells (Figure 1A,B). Interestingly, we observed a significant decrease in H3R26 symmetric arginine dimethylation in DNA-damaged cells, while H3R26 asymmetric arginine dimethylation showed no significant change (Figure 1A,B,D). This suggests distinct functions for symmetric and asymmetric histone arginine methylation in response to DNA damage. Additionally, it implies potential differences in the chromatin locations of H3R26me2s and H3R26me2a. Similar alterations in histone H3R26 methylation were observed in Hela cells (Figure 1C,D), and consistent trends in H3R26 methylation were visualized using immunofluorescence in HepG2 cells (Figure 1E–G). Notably, the decreased ratio of H3R26me2s levels varied among the three cell lines, indicating that cell lines have distinct sensitivities to BLM. These findings suggest that the H3R26 methylation pattern represents a general cellular response to stress.

To validate the modification changes in the H3R26 site with a range of BLM concentrations or with a series of time-point treatments, we treated HepG2 cells with different concentrations of BLM (5 μg/mL, 10 μg/mL, and 20 μg/mL) for 24 and 48 h. As depicted in Figure 2A, H3R26me2s levels progressively decreased with increasing BLM concentration and treatment duration (Figure 2A,B). A549, 293, and HFF cell lines were treated with different BLM concentrations for 24 h, and the result was decreased H3R26me2s levels (Appendix A). However, the reduction ratio may not be consistent with an increase in BLM concentration, suggesting that a low BLM concentration was sufficient to induce a reduction in H3R26me2s. Finally, treatment of A549 cells with 10 μg/mL BLM for 6, 24, 48, 72, and 96 h resulted in a gradual decrease in H3R26me2s levels with increasing treatment time (Appendix A). Notably, in Appendix A, H3R26me2s and H3K27ac reduction levels did not correct with γH2A.X—which reflects damage accumulation—level increase, indicating that histone modification changes may not tightly be following DNA damage response.

To determine whether these dynamic patterns of histone modification were prevalent under stress induced by other chemical reagents, we exposed HepG2 cells to H_2_O_2_ or temozolomide (TMZ) to induce DNA damage. Cells were treated with 1 mM H_2_O_2_ for 1, 2, 4, or 6 h. H3R26me2s, but not H3R26me2a, decreased in a time-dependent manner (Figure 2C,D). In the TMZ treatment group, HepG2 cells treated with 100 μM, 200 μM, and 400 μM TMZ for 24 h showed a decrease in H3R26me2s levels with increasing TMZ concentrations, while H3R26me2a remained unchanged (Figure 2E,F). These results suggest that changes in symmetric histone arginine methylation at H3R26 are a common cellular response to stress induced by chemical treatment.

Considering the observed changes in H3R26me2s in response to stress-induced DNA damage, we investigated the genome-wide distribution of these modifications. We first analyzed the changes in γH2A.X peaks in BLM-treated cells and found that over 90% of γH2A.X peaks increased (Appendix A). Calculating the number and fold change in histone modification peaks revealed a decrease in the median total of H3R26me2s, while γH2A.X was enriched as expected (Figure 1H). Further analysis of the genomic loci of the H3R26me2s peaks revealed that over 50% of the peaks were predominantly located within the gene body, approximately 28% were interspersed intergenically, and a smaller fraction (approximately 11%) was found in the promoter region (Figure 1I). This suggests that the H3R26me2s peaks are mainly located near genes and may participate in gene regulation. Genes near the H3R26me2s peaks were annotated and analyzed using topGO (v2.42.0) software to determine their Gene Ontology (version 24.1.1). The analysis revealed that genes linked to the reduced H3R26me2s peaks in BLM-treated cells were mainly involved in chromatin maintenance, nucleic acid metabolic processes, and metabolic processes (Figure 1G). These genes may play a role in DNA damage response related to histone arginine methylation.

#### 2.1.2. H3K27ac Undergoes Hypoacetylation in Response to Stress-Induced DNA Damage

Given that H3K27 is adjacent to H3R26, we investigated whether alterations in H3R26 methylation could affect H3K27 modifications under stress-induced DNA damage. We found that BLM-induced DNA damage resulted in decreased H3K27ac levels in HepG2, LM3, and Hela cells without significant changes in H3K27me3 levels (Figure 3A–D). These findings were consistent with previous reports [29,30] and were visualized using immunofluorescence (Figure 3E,F).

The change in H3K27ac levels was validated by treating HepG2 cells with varying BLM concentrations or a time course of BLM treatment. A gradual reduction in H3K27ac was observed with increasing BLM concentrations or treatment times, while H3K27me3 levels remained stable (Figure 2A,B). In A549, 293, and HFF cell lines, the level of H3K27ac also decreased with increasing BLM concentrations; however, the extent of reduction varied among cell lines, even with the same concentration of BLM (Appendix A). In A549 cells, H3K27ac levels gradually decreased with increasing BLM treatment time (Appendix A).

To assess if these dynamic patterns of histone modifications were prevalent under stress induced by other chemical reagents, we examined H3K27ac and H3K27me3 levels in HepG2 cells treated with H_2_O_2_ and TMZ. In H_2_O_2_-induced DNA-damaged cells, we observed a decrease in H3K27ac levels, while H3K27me3 levels remained unchanged (Figure 2C,D). In the TMZ-induced DNA damage model, we detected a decrease in H3K27ac levels, with no change in H3K27me3 levels (Figure 2E,F). These results indicate that deacetylation of the H3K27 site is conserved in response to chemical reagent stress.

Previous studies have shown that HDACs are essential in NHEJ during DNA damage. In this study, we treated DNA-damaged cells with an HDAC inhibitor (HDACi) to determine if the reduction in H3K27ac could be blocked. The levels of H3R26me2a and H3K27me3 remained unchanged in the BLM and BLM+HDACi groups compared with the control group. H3R26me2s levels were slightly lower in the BLM+HDACi group than in the BLM group. The BLM+HDACi group effectively reversed the hypoacetylated state of H3K27 (Figure 3G,H), indicating HDAC involvement in the deacetylation process of H3K27ac. It has been reported that HDAC expression is not significantly altered by DNA damage [31]. HepG2 cells were treated with 10 μg/mL or 20 μg/mL for 24 h to investigate whether HDAC1 translocated into the nucleus after DNA damage. HDAC1 expression showed a 20% increase in the BLM group compared to the control group (Figure 3I). Cytoplasmic and nuclear contents were isolated, and HDAC1 levels were detected. In the cytoplasm, HDAC1 decreased in the BLM group, while in the nucleus, HDAC1 also decreased in the BLM group (Figure 3J,K).

Subsequently, we analyzed the changes in H3K27 site modifications in BLM-treated cells. We found approximately 894 and 148 peaks with decreased and increased H3K27ac, respectively, in BLM-treated cells. Although there was no global change in H3K27me3 levels, its peaks changed in many areas of the genome, with approximately 165 decreasing and 495 increasing peaks (Appendix A). Calculation of the number and fold changes in histone modifications at different peaks showed an overall decrease in H3K27ac levels, with minimal changes in H3K27me3 levels. These findings are consistent with the immunoblotting assay results (Appendix A). Additionally, genes located near different peaks of H3K27ac regulated transcription (Appendix A).

#### 2.1.3. H3K27 Acetylation or Trimethylation Does Not Modulate H3R26me2s Levels

Our study revealed a decrease in H3R26me2s and H3K27ac levels during DNA damage, prompting an investigation into their interactive regulation. First, HepG2 cells were treated with a P300 inhibitor; as anticipated, the level of H3K27ac decreased, while the level of H3K27me3 remained unaffected, with no change in H3R26me2s levels (Figure 4A,B). Subsequently, HepG2 cells were treated with the HDAC inhibitors LHB598 or trichostatin A (TSA), leading to an increase in H3K27ac levels while H3R26me2s levels remained unchanged (Figure 4C–F). These results suggest that changes in H3K27ac do not induce changes in H3R26 symmetric methylation. A previous study introduced an H3R26A mutation in *Drosophila melanogaster* [1], which significantly reduced the level of H3K27me3, indicating a relationship between H3R26 and H3K27me3. HepG2 cells were treated with the enhancer of zeste 2 polycomb repressive complex 2 (EZH2) inhibitor DZNep, to investigate whether H3K27 methylation affects H3R26 methylation. H3K27me3 levels decreased with DZNep treatment, but the reduction level was similar in both the 5 μM and 10 μM DZNep treatment groups. The level of H3R26me2s did not change in cells treated with DZNep, while H3K27ac levels increased slightly (Figure 4G,H). These results suggest that H3K27me3 demethylation does not impact the state of H3R26me2s.

#### 2.1.4. H3R26me2s Modulate H3K27 Acetylation State

PRMT5 and PRMT9 have been reported to catalyze arginine symmetric dimethylation. To verify their role in catalyzing H3R26me2s, we used siRNA to knock down *PRMT5* and *PRMT9* in HepG2 cells and detected changes in H3R26me2s levels. The expression levels of *PRMT5* and *PRMT9* decreased in *siPRMT5* and *siPRMT9* transfected cells (Figure 5A,B). Modifications at the H3R26 and H3K27 sites were detected in these cells. The level of H3R26me2s was decreased to 70% in *PRMT5* knockdown cells, but no change was observed in *PRMT9* knockdown cells (Figure 5C). These results suggest that PRMT5 may be responsible for H3R26me2s methylation. We also analyzed the efficiency of PRMT5 binding to chromatin. Although CUT&Tag-seq acquired a lower number of peaks, we found that 52 peaks decreased and only two peaks increased compared to the control group (Figure 5D,E). These results suggest that PRMT5 dissociates from the chromatin in BLM-treated cells, resulting in histone arginine demethylation.

Treatment of cells with the PRMT5 inhibitor GSK591 reduced H3R26me2s levels to 56% and 18% with 10 μM and 20 μM GSK591 treatment, respectively (Figure 5F). Additionally, H3K27ac levels decreased, while H3K27me3 levels remained unchanged (Figure 5F). These results suggest that H3R26 methylation mediates H3K27 acetylation. To validate these findings, we generated a stable *PRMT5* knockdown cell line in LM3 cells. In these cells (Figure 5G), H3R26me2s levels decreased, while levels of H3R26me2a remained unchanged. As expected, the H3K27ac levels decreased with a modest increase in H3K27me3 levels (Figure 5G,H). A recent study supports our findings, demonstrating that *PRMT5* knockout increases H3K27me3 levels [32]. PRMT5 has been reported to indirectly impair the function of PRC2, indicating a complex modulation. We did not observe any changes in the levels of H3K27me3 in siRNA knockdown PRMT5 HepG2 cells (Figure 5C), possibly due to the use of different cell lines.

Since PRMT5 is primarily distributed in the cytoplasm, we fused a nuclear localization signal (NLS) peptide to its N-terminus and transfected it into HepG2 cells (Figure 5I). After 48 h of transfection, we measured changes in histone modifications via immunoblotting. The results revealed increased H3R26me2s and a slight increase in H3R26me2a, elevated H3K27ac, and unchanged H3K27me3 (Figure 5I,J). Therefore, alterations in H3R26me2s and H3K27ac observed in cells with PRMT5 knockdown as well as overexpression provide additional evidence supporting our hypothesis that histone arginine methylation affects lysine acetylation.

#### 2.1.5. H3R26me2s Demethylation Recruit HDAC1 to Mediate H3K27 Deacetylation

H3R26me2s mediates H3K27ac changes, and, during DNA damage, HDAC1 is translocated into the nucleus. Therefore, we hypothesize that H3R26me2s demethylation may recruit HDAC1, which in turn catalyzes H3K27 deacetylation.

We analyzed changes in H3R26me2s and H3K27ac over time in HepG2 and A549 cells treated with BLM or H_2_O_2_. In A549 cells, H3R26me2s expression decreased earlier than that of H3K27ac (Appendix A and Figure 6A). Similarly, in HepG2 cells, H3R26me2s showed an early decrease compared to H3K27ac (Figure 2C and Figure 6B). These results suggest that H3R26me2s demethylation occurs earlier than H3K27ac deacetylation.

To validate HDAC1 binding bias, we used the *glyceraldehyde-3-phosphate dehydrogenase* (GAPDH) gene locus as a model (Figure 6C). Cleavage under targets and tagmentation (CUT&Tag) qPCR was used to detect HDAC1 binding efficiency in the H3R26me2s high region (locus #1) and the H3R26me2s low region (locus #2) of HepG2 cells. We found that in the H3R26me2s lower region, HDAC1 also showed a lower enrichment (Figure 6D). Given that the H3R26me2s lower region overlaps with the H3K27ac lower region, to eliminate the possibility of HDAC1 catalyzing H3K27ac deacetylation and mediating H3R26 demethylation, we overexpressed HDAC1 in HepG2 cells (Figure 6E). Subsequently, we analyzed the H3K27ac, H3R26me2s, PRMT5, and HDAC1 levels at the *lysophosphatidylcholine acyltransferase 1* (LPCAT1) gene locus (Figure 6F). We observed an increase in HDAC1 binding efficiency and a decrease in H3K27ac levels. However, there was no change in the fold enrichment of PRMT5 or H3R26me2s (Figure 6G). These results reveal that HDAC1 mediates H3K27 deacetylation and did not impair the H3R26me2s state. In conclusion, our results suggest that H3R26me2s deacetylation may serve as a bridge for H3K27ac deacetylation by recruiting HDAC1.

#### 2.1.6. Histone H3R26me2s Co-Localizes with H3K27ac throughout the Genome

To investigate H3R26me2s and H3K27ac distribution features in the genome, HepG2 cells were treated with BLM to induce DNA damage. Histone modification information was acquired using CUT&Tag combined with next-generation sequencing [33,34,35,36].

First, we analyzed the distribution of these histone modifications. As expected, the H3K27ac and H3K27me3 modifications were largely antagonistic across the genome. Interestingly, H3R26me2s exhibited similar occupancy to H3K27ac across chromatin and antagonized H3K27me3 (Figure 7A,B). This distribution pattern suggests that H3R26me2s and H3K27ac have similar functions. Notably, H3R26me2a also showed co-occupancy with H3R26me2s in the euchromatin region; however, the peak density of H3R26me2a was lower than that of H3R26me2s. Nevertheless, CUT&Tag-seq suggested that fewer than ten peaks of H3R26me2a were altered in the BLM group, indicating that H3R26me2a and H3R26me2s have distinct functions and distributions in chromatin.

To validate the relationship between H3R26me2s and H3K27ac, we analyzed their overlapping peaks. Over 70% of these peaks exhibited the same pattern of change across the genome (Figure 7C), further confirming crosstalk in response to DNA damage. Genes associated with these regions are involved in RNA processing, gene expression, and cellular metabolic processes (Figure 7D). These results overlapped with different peaks associated with H3R26me2s and H3K27ac (Figure 1J and Appendix A).

To assess the effect of these regions on gene expression, we performed RNA-seq analysis of gene expression in DNA-damaged and normal cells. Our findings revealed that 256 genes were upregulated and 84 were downregulated (Figure 7E,F). The downregulated genes were primarily associated with cellular metabolic processes, whereas the upregulated genes were associated with type I IFN or antiviral processes and RNA transcription (Appendix A). In the set of upregulated genes, including *growth arrest and DNA damage inducible alpha* (GADD45A) and *interferon stimulated exonuclease gene 20* (ISG20), elevated levels of their promoters, H3R26me2s and H3K27ac (Figure 8A,B), were observed. The *GADD45A* gene was reported to serve as an important factor that participates in cellular response to a variety of DNA damage agents [37], indicating that H3R26me2s participates in the DNA damage response. Conversely, the downregulation of *LPCAT1* was associated with a decrease in H3R26me2s and H3K27ac (Figure 7G). However, unlike *GADD45A* and *ISG20*, H3R26me2s and H3K27ac were distributed within the *LPCAT1* gene body (Figure 7H). Therefore, H3R26me2s may play a role in the regulation of gene expression in response to DNA damage stress.

## 3. Discussion

PRMTs were reported to participate in the DNA damage response; PRMT1 methylated non-histone protein MRE11 and 53BP1 maintain genomic stability [38]. PRMT5-deficient cell lines were sensitive to DNA damage [39,40] PRMT5 also controls the H4Kac level to regulate DNA damage repair by mediating TIP60 splicing [41]. PRMT7 was reported to catalyze H2AR3 and H4R3 site methylation and negatively regulate DNA damage repair genes; knockdown of PRMT7 could derepress these genes expression and promote DNA repair [42]. However, how the PRMTs catalyze the histone arginine residue function is largely unknown. In this study, we first found that in the histone H3R26 site, symmetric methylation was decreased in DNA-damaged cells; furthermore, H3R26me2s decrease is conserved in multi-cell lines. We found decreased symmetric arginine demethylation at histone H3R26, while asymmetric arginine demethylation at the same site remained unchanged. The symmetric and asymmetric methylation differences are not limited to gene regulation but expand to the DNA damage response. However, the reason for PRMT5 in response to DNA damage signals and dissolution from chromatin is still unknown, indicating a need for further research to understand its mechanism of dissociation from chromatin. Our results found that PRMT5 was dissolved from chromatin, but which signal pathway promoted this still needs to be studied. Moreover, the histone arginine demethylation mechanism is not fully understood, two proteins were reported as histone arginine demethylase: JMJD6 and PADI4 [43,44]. However, the following study demonstrated that JMJD6 was a lysine hydroxylase and unable to detect demethylase activity on either H3R2me2 or H4R3me2 peptides [45]. PADI4 could catalyze the deimination of arginine, but not methylated arginine residues, and can block methylation on an arginine residue [44]. Whether histone arginine demethylase may participate in H3R26me2s demethylation may be a promising area to explore. 

Histone H3K9ac and H3K56ac were reported to undergo deacetylation in the DNA damage response [29]. Inhibition of HDAC results in abnormalities in the HDR and NHEJ DNA break repair pathway [46]. In our study, we found that H3K27ac was also decreased in response to DNA damage; HDAC1 was translocated into the nucleus during DNA damage and mediated H3K27ac deacetylation. Previous studies demonstrated that H4R3me2a is a P300 bias binding site [47]; because demethylation of H3R26me2s occurs first in response to DNA damage, and HDAC1 is translocated into the nucleus, we found that HDAC1 was recruited to these chromatin regions to mediate H3K27ac deacetylation. These results indicate that lysine deacetylation of other sites may also be regulated by adjacent arginine demethylation. However, it remains unknown whether other HDACs participate in this process. Additionally, further evaluation is required to determine whether HDAC1 directly binds to the lower region of H3R26me2s. On the other hand, whether H3R26me2s demethylation may prevent lysine acetylase or recruit lysine methylase is also important to explore in further investigation. Histone lysine modifications were reported to recruit NHEJ or HR pathway factors to facilitate DNA repair [48]. H3R26me2s dynamic changes in DNA damage not only recruit HDAC1 but also act as an important site for DNA break site repair.

The histone lysine modifications that have been well studied include the site of H3K27, H3K36, H3K9, and H3K4 acetylation or methylation, and were evaluated by ENCODE and the Roadmap project, including their distribution pattern and function in regulating gene expression or chromatin; these resources are helpful for lysine modification research [49,50]. However, the arginine methylation distribution data are rarely studied. Using CUT&Tag in combination with next-generation sequencing, we obtained information on the distribution of histone modifications across the genome, we first depicted the H3R26me2s distribution feature, and H3R26me2s have a distribution pattern similar to that of H3K27ac. H3K27ac acts as an enhancer or promoter activity marker, due to the H3R26me2s modulating the H3K27ac model, H3R26me2s may also be an important epigenetic marker of activated enhancers. Gene expression analysis revealed that the promoters of the upregulated genes showed both H3R26me2s and H3K27ac modifications, suggesting that H3R26me2s may influence gene expression. These results indicate that histone arginine methylation, in combination with adjacent lysine modifications, modulates gene expression. H3K27ac or H3K4me1/me3 is always used for predicate gene expression and chromatin state [51], but, using the dCas9 system for gene-specific activity, H3K27ac increased promoter or enhancer did not promote gene expression [52,53]. A dCas9 system plus PRMTs for gene regulation may be a promising method.

Although H3R26me2s co-localizing with H3K27ac was proved in our studies, we only analyzed the CUT&Tag-seq data in HepG2 cell lines; a broad cell lines or in vivo sample is needed for H3R26me2s enrichment analysis; the lack of arginine ChIP-seq data limits further investigation. A project on the analysis of histone arginine methylation feature may be needed. 

Figure 9 summarizes the biological changes in response to DNA damage stress. Our research identified the distribution and dynamic changes in H3R26 methylation and its crosstalk with adjacent H3K27 acetylation. These findings provide a new perspective regarding histone arginine methylation.

## 4. Materials and Methods

### 4.1. Cell Culture and Treatment

HEK293T, Hela, HepG2, and LM3 cells were cultured in DMEM/High glucose supplemented with 10% FBS, 1% Penicillin–Streptomycin solution at 37 °C in 5% CO_2_. For bleomycin (sellect)-induced DNA damage, cells at 60% confluency were treated with 10 μg/mL bleomycin for 24 h. For H_2_O_2_-induced DNA damage, cells at 80% confluency were treated with 1 mM H_2_O_2_ for 1~6 h. Detailed information on cell lines used in this study can be seen in Appendix A.

For P300 inhibitors 29-2 and 85-2, inhibitors were dissolved with DMSO, and cells were treated with 0.5 μM inhibitors for 48 h. DZNep (sellect) was dissolved with DMSO and used to treat cells at quantities of 5 μM and 10 μM for 72 h. HDAC inhibitor LBH589 (sellect) was dissolved with DMSO and was used to treat cells at 0.01 μM for 24 h. HDAC inhibitor TSA (sellect) was dissolved with DMSO and used to treat cells at quantities of 0.01 μM, 0.05 μM, and 0.2 μM for 24 h. PRMT5 inhibitor GSK591 (sellect) was dissolved with DMSO and used to treat cells at 5 μM, 10 μM, and 20 μM for 48 h.

### 4.2. Generate Stable Knockdown Cell Lines

Lentiviral pLKO.1-based shRNA plasmid was used to generate knockdown cell lines. pLKO.1, pSPAX2, and pMD2G plasmids were co-transfected into HEK293T cells for 48–72 h, and the supernatant was collected and filtered with a 0.45 μM filter. LM3 cells were infected with lentiviral for 72 h and selected with 1 μg/mL puromycin. All stable knockdown cell lines used in our research were mono-cloned cell lines. The shRNA sequences used in this study are as follows: 

shPRMT5: 5’- CCCATCCTCTTCCCTATTAAG -3’

### 4.3. CUT&Tag

The CUT&Tag method was performed as previously described [33]. Briefly, the ProtenA/G-Tn5 expression plasmid was transformed into BL21 (DE3). Then, one clone was picked into 50 mL LB (containing 100 μg/mL ampicillin) and grown overnight at 37 °C; 20 mL of cultured cells were transferred into 1 L LB (containing 100 μg/mL ampicillin) and cultured at 37 °C until the O.D. value reached 0.6~0.8. Fresh IPTG was added to the medium through a 0.2 μM filter to induce ProteinA/G-Tn5 expression. The culture was incubated for 6 h at 23 °C, and bacteria were then collected via centrifugation and resuspended with HEGX buffer. The lysate was sonicated and centrifuged at 12,000 rpm for 10 min at 4 °C; then, the soluble fraction was mixed with chitin slurry resin (NEB, S6651S) overnight, and the unbound soluble was discarded via centrifugation. The rest of the chitin slurry resin was mixed with solution buffer (HEGX buffer with 100 mM DTT) and rotated for 48 h at 4 °C. After centrifugation, the solution was collected and concentrated through an ultracentrifuge column (Millipore); purified protein was then stocked by mixing with glycerol to a final concentration of 50%.

To generate the active transposome, 16 μL of 100 μM adaptor was mixed with 100 μL 5.5 μM proteinA/G-Tn5 fusion protein; the transposome was incubated on a rotator at room temperature for 1 h and stored at −20 °C.

To generate histone modification information, 100,000 cells were used per sample. Cells were harvested and washed with a wash buffer (20 mM HEPES pH 7.5, 150 mM NaCl, 0.5 mM Spermidine, and 1× Protease inhibitor cocktail). Concanavalin A-coated magnetic beads (Bangs Laboratories) were washed with 1mL binding buffer (20 mM HEPES pH 7.5, 10 mM KCl, 1 mM CaCl_2_, 1 mM MnCl_2_) and then resuspended in the binding buffer; each sample used 10 μL ConA beads. Then, 10 μL of ConA beads were added to the cell sample and rotated at room temperature for 10 min; cells were collected by placing the tubes on the magnet stand. Cells were resuspended in 200 μL antibody buffer (wash buffer with 0.01% digitonin, 2 mM EDTA, and 0.1% BSA) with 1 μL antibody of interest, then incubated overnight at 4 °C. The antibody buffer was removed, and cells were washed with dig-wash buffer twice; then, the secondary antibody was added in 200 μL dig-wash buffer and incubated at room temperature for 1 hour.

The buffer was removed and cells were washed with dig-wash buffer twice; cells were then resuspended in dig-300 buffer (0.05% Digitonin, 20 mM HEPES, pH 7.5, 30 mM NaCl, 0.5 mM Spermidine, 1× Protease inhibitor cocktail). The transposome was added to the buffer in a 1:100 dilution manner, then incubated at room temperature for 1 h. The buffer was removed and cells were washed with dig-300 buffer twice; then, cells were resuspended with 300 μL tagmentation buffer (dig-300 buffer with 10 mM MgCl_2_) and incubated at 37 °C for 1 hour. To stop tagmentation, 10 μL 0.5 M ETAD, 3 μL 10% SDS, and 2.5 μL 20 mg/mL Proteinase K were added and incubated at 56 °C for 1 hour. The DNA library was constructed using NEB Next HiFi 2× PCR Mix, and the PCR products were purified with VAHTS DNA clean beads (Vazyme). 

### 4.4. Fraction

Cultured cells were collected and washed with 1× PBS; cell pellets were resuspended in 200 μL cold cytoplasmic lysis buffer (0.15% NP-40,10 mM Tris-HCl pH 7.5, 150 mM NaCl) using wide orifice tips and incubated on ice for 5 min. The lysate was layered onto 500 μL cold sucrose buffer (10 mM Tris pH7.5, 150 mM NaCl, and 24% sucrose *w*/*v*), and centrifuged in microfuge tubes at 13,000 rpm for 10 min at 4 °C. The supernatant from this spin (700 μL) represented the cytoplasmic fraction. The nuclear pellet was gently resuspended into 200 μL RIPA buffer with 1 mM PMSF and 1× Protease inhibitor cocktail.

### 4.5. Western Blot

Cells were lysed with RIPA buffer with 1 mM PMSF and 1× Protease inhibitor cocktail; protein concentration was quantified using a BCA assay, and 10 μg of total protein was dissolved with 10% or 15% SDS-PAGE and transferred electrically to PVDF membranes. The PVDF membranes were blocked with 3% BSA with PBS plus 0.1% tween-20, then incubated with primary and secondary antibody. Detailed information on antibodies used in this study can be seen in Appendix A.

### 4.6. Immunofluorescence Assays

Cells were grown on glass coverslips; after transfection with chromatins or treated with bleomycin, cells were washed with PBS three times, then fixed with 4% paraformaldehyde at room temperature for 15 min and washed with PBS, followed by permeabilization with 0.2% Triton X-100 at room temperature for 10 min and then washed with PBS; then, cells were blocked with 3% BSA at room temperature for 30 min. Then, cells were incubated with the antibodies of interest overnight at 4 °C. Cells were washed with PBS three times and incubated with secondary antibody Alexa Flour 594 IgG at room temperature for 1 h, then washed again with PBS. Nuclei were stained with 5 μg/mL DAPI at room temperature for 5 min and then washed with PBS; cells were covered with 50% glycerol for use.

### 4.7. Cell Transfection

Cells were cultured to 70–80% confluency, then transfected with the plasmid, siRNA, or chromatin via Hieff Trans^TM^ liposome nucleic acid transfection reagent (YEASEN) following the manufacturer’s instruction. The medium was changed at 6–8 h after transfection. Cells were harvested at 48 h after transfection to detect the gene expression using qPCR or Western blot.

### 4.8. CUT&Tag Data Analysis

First, we removed the adaptor sequence, the files of R1 and R2 were quantified with DynamicsTrim [34], and then the data were mapped to the hg38 genome using bowtie2 software [35] and PCR duplicated sequences were removed. Enriched peaks were collected using the mapped data, and the bleomycin treatment and normal cell peaks were analyzed to compare their different regions. The different enrichment peaks were annotated using DAVID software version 6.8 (last update in 2022) [36], and functional enrichment of nearby genes was analyzed using Gene Ontology (version 24.1.1) and KEGG (version 109.1).

## Figures and Tables

**Figure 1 ijms-25-07562-f001:**
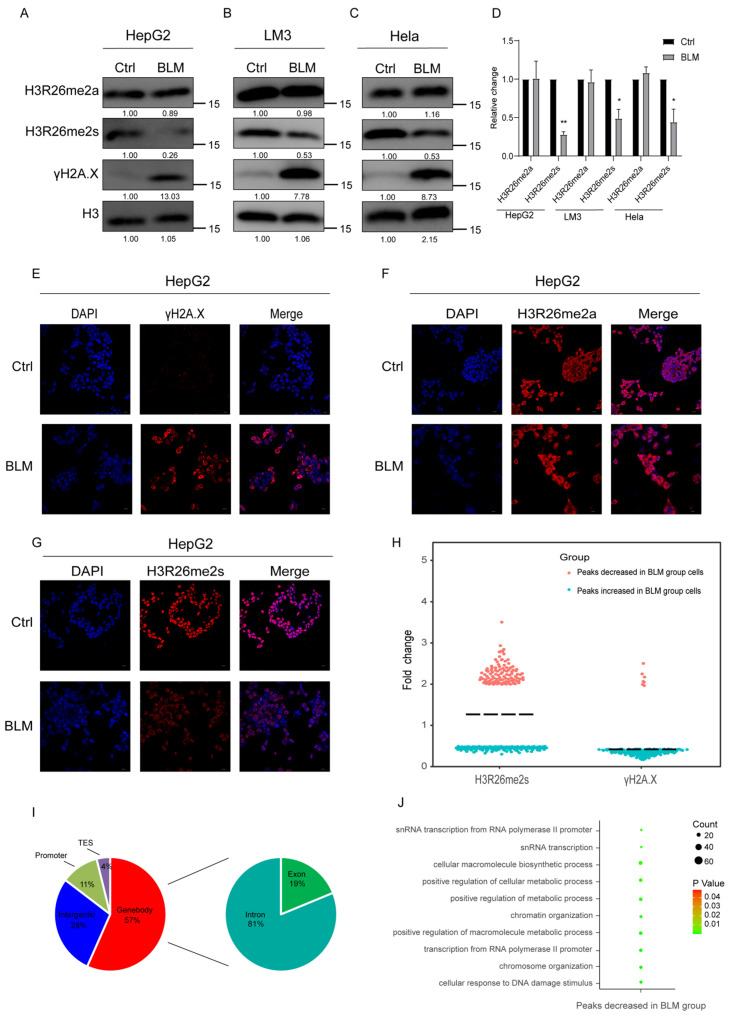
Histone arginine methylation decreased in DNA damaged cells. (**A**) H3R26me2s methylation level was decreased in bleomycin-treated HepG2 cells analyzed using immunoblot assay, while for H3R26me2a there was no change. (**B**) H3R26me2s methylation level was decreased in bleomycin-treated LM3 cells analyzed using immunoblot assay, while for H3R26me2a there was no change. (**C**) H3R26me2s methylation level was decreased in bleomycin-treated Hela cells analyzed using immunoblot assay, while for H3R26me2a there was no change. (**D**) Relative changes in H3R26me2a and H3R26me2s in bleomycin-treated HepG2 (**A**), LM3 (**B**), and Hela (**C**) cells. (**E**) γH2A.X modification was increased in bleomycin-treated HepG2 cells measured using immunofluorescence. (**F**) Immunofluorescence of H3R26me2a in control and bleomycin-treated cells. (**G**) Immunofluorescence of H3R26me2s in control and bleomycin-treated cells. (**H**) Nearly half of the H3R26me2s modification peaks were downregulated in bleomycin-treated cells, as a positive control, 93% percent γH2A.X of peaks were upregulated in bleomycin-treated cells. (**I**) Percentage of H3R26me2s’s different peaks’ distribution features across the genome. Left Pie Chart: Characteristics of H3R26me2s’ different peaks in genome features of Genebody, Intergenic, Promoter, and TES regions. Right Pie Chart: Percentage of H3R26me2s different peaks distribution in Exon and Intron regions. (**J**) Analysis of H3R26me2s different peaks associated with gene functions using Gene Oncology. Data information: Scale bars in micrographs = 10 μm. Data are shown as the mean ± SEM of three biological replicates and were analyzed with a two-tailed unpaired Student’s *t*-test. *: *p* < 0.05, **: *p* < 0.01.

**Figure 2 ijms-25-07562-f002:**
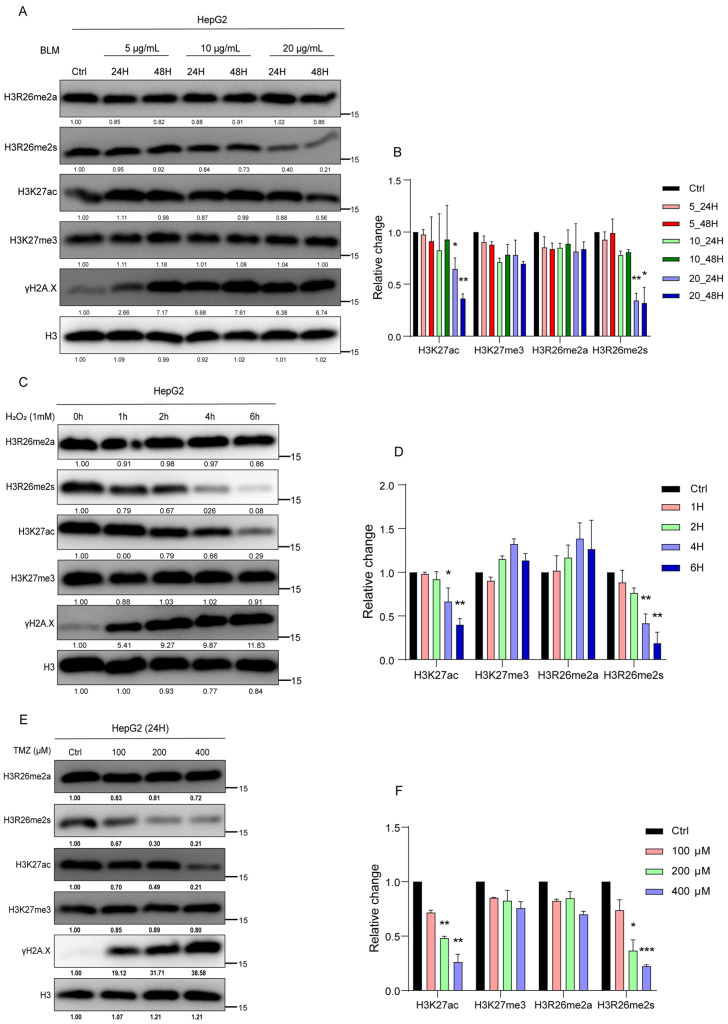
Histone modifications respond to the stress of chemical reagents in HepG2 cells. (**A**) HepG2 cells were treated with 5 μg/mL, 10 μg/mL, and 20 μg/mL bleomycin; for each BLM concentration group, cells were collected at 24 h and 48 h. Histone modifications were detected using immune blot with the indicated antibodies. (**B**) Relative changes in H3R26me2a, H3R26me2s, H3K27ac, and H3K27me3 in bleomycin-treated HepG2 cells (**A**). (**C**) HepG2 cells were treated with 1 mM H_2_O_2_ for 1, 2, 4, and 6 h, cell lysate was collected, and histone modifications were detected with the indicated antibodies. (**D**) Relative changes in H3R26me2a, H3R26me2s, H3K27ac, and H3K27me3 in H_2_O_2_-treated HepG2 cells (**C**). (**E**) HepG2 cells were treated with 100 μM, 200 μM, and 400 μM TMZ for 24 h, cell lysate was collected, and histone modifications were detected with the indicated antibodies. (**F**) Relative changes in H3R26me2a, H3R26me2s, H3K27ac, and H3K27me3 in TMZ-treated HepG2 cells (**E**). Data are shown as the mean ± SEM of two biological replicates and were analyzed with a two-tailed unpaired Student’s *t*-test. *: *p* < 0.05, **: *p* < 0.01, ***: *p* < 0.001.

**Figure 3 ijms-25-07562-f003:**
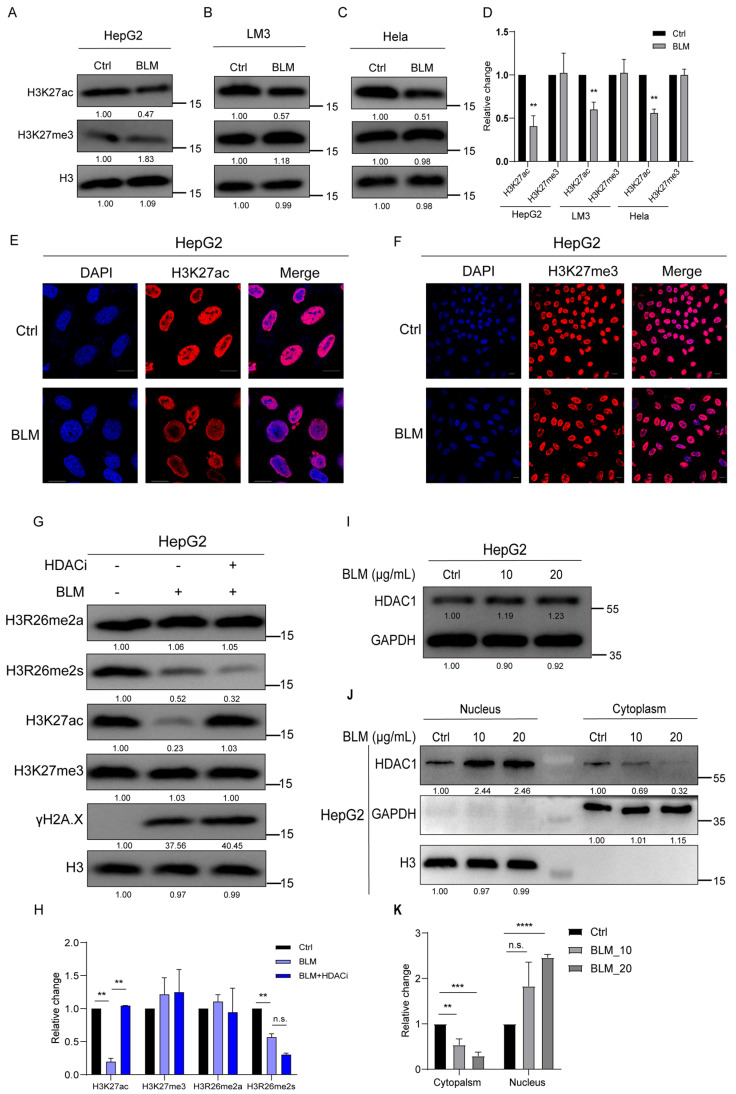
H3K27ac undergoes hypoacetylation in response to DNA damage. (**A**) H3k27ac level was decreased in bleomycin-treated HepG2 cells analyzed using immunoblot assay, and the H3K27me3 level showed no change. (**B**) H3k27ac level was decreased in bleomycin-treated LM3 cells analyzed using immunoblot assay; H3K27me3 level showed no change. (**C**) H3k27ac level was decreased in bleomycin Hela cells analyzed using immunoblot assay; H3K27me3 level showed no change. (**D**) Relative changes in H3K27ac and H3K27me3 in bleomycin-treated HepG2 (**A**), LM3 (**B**), and Hela (**C**) cells. (**E**) Immunofluorescence of H3K27ac in control and bleomycin-treated cells; H3K27ac was decreased in bleomycin-treated cells. (**F**) Immunofluorescence of H3K27me3 in control and bleomycin-treated cells; H3K27me3 showed no change in bleomycin-treated cells. (**G**) H3K27ac level was reversed in treatment with bleomycin and HDAC inhibitor LBH598 cells. (**H**) Relative changes in H3K27ac, H3K27me3, H3R26me2a, and H3R26me2s in BLM- or BLM+HDACi-treated HepG2 cells (**G**). (**I**) Immune blot analysis of HDAC1 expression in bleomycin-treated HepG2 cells. (**J**) HDAC1 was detected in the cellular nucleus and cytoplasmic content using immunoblot; H3 and GAPDH were used as a nucleus or cytoplasm marker. (**K**) Relative levels of HDAC1 in cytoplasmic or nucleus in bleomycin-treated HepG2 (**J**) cells. Data information: Scale bars in micrographs = 10 μm. Data were analyzed with a two-tailed unpaired Student’s *t*-test. **: *p* < 0.01, ***: *p* < 0.001, ****: *p* < 0.0001, n.s.: no significance.

**Figure 4 ijms-25-07562-f004:**
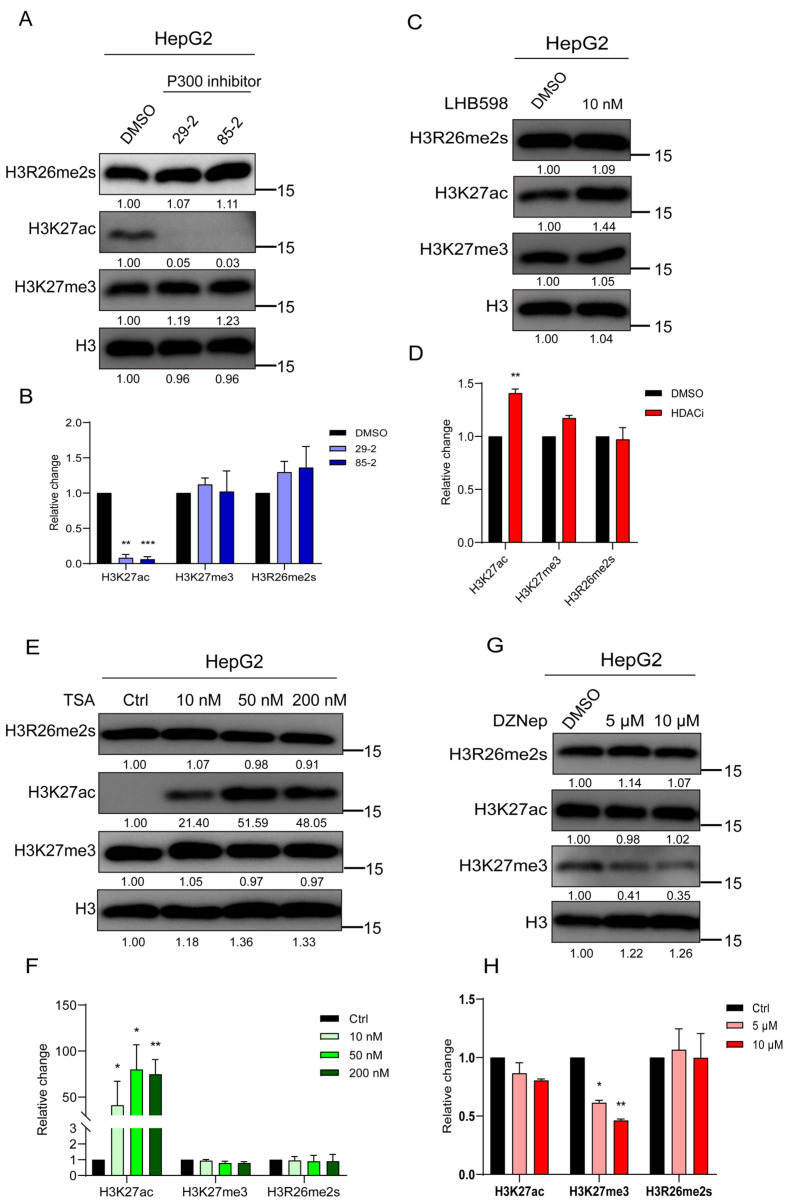
H3K27 acetylation or trimethylation does not impair H3R26me2s. (**A**) Cells were treated with P300 inhibitors for 48 h to inhibit histone lysine acetylation, and the H3R26 methylation state detected using immunoblot assay was not impaired. (**B**) Relative changes in H3R26me2s, H3K27ac, and H3K27me3 in P300 inhibitor-treated HepG2 cells (**A**). (**C**) Cells were treated with 10 nM HDAC inhibitor LHB598 for 24 h to evaluate histone lysine acetylation, the H3R26 symmetric dimethylation state detected via immunoblot assay was not impaired. (**D**) Relative changes in H3R26me2s, H3K27ac, and H3K27me3 in HDAC inhibitor LHB598-treated HepG2 cells (**C**). (**E**) Cells were treated with 10 nM, 50 nM, and 200 nM HDAC inhibitor TSA for 24 h to evaluate histone lysine acetylation; the H3R26 symmetric dimethylation state detected via immunoblot assay was not impaired. (**F**) Relative changes in H3R26me2s, H3K27ac, and H3K27me3 in HDAC inhibitor TSA-treated HepG2 cells (**E**). (**G**) Cells were treated with 5 μM and 10 μM EZH2 inhibitor DZNep for 72 h to inhibit histone lysine methylation, the H3R26 methylation state detected via immunoblot assay was not impaired. (**H**) Relative changes in H3R26me2s, H3K27ac, and H3K27me3 in EZH2 inhibitor DZNep-treated HepG2 cells (**G**). Data information: Data were analyzed with a two-tailed unpaired Student’s *t*-test. *: *p* < 0.05, **: *p* < 0.01, ***: *p* < 0.001.

**Figure 5 ijms-25-07562-f005:**
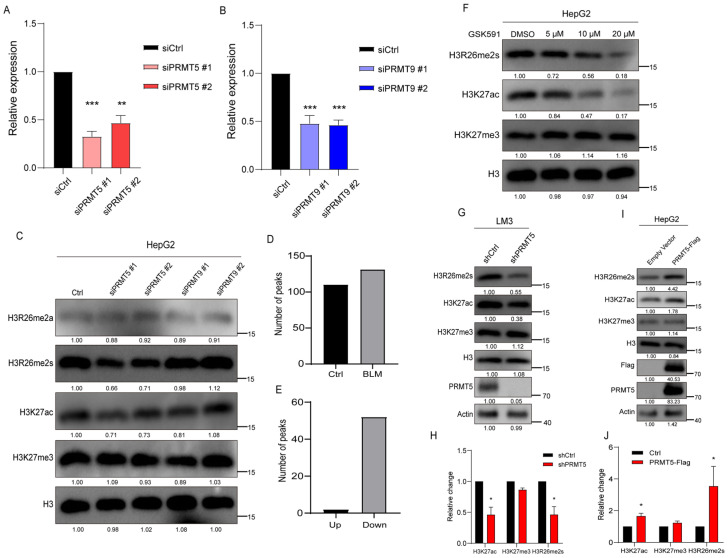
Histone H3R26me2s modulates H3K27 acetylation state. (**A**) qPCR analysis of *PRMT5* knockdown efficiency by siRNA in HepG2 cells. (**B**) qPCR analysis of *PRMT9* knockdown efficiency by siRNA in HepG2 cells. (**C**) *PRMT5* and *PRMT9* were knocked down in HepG2 cells by siRNA, and histone modifications were detected with the indicated antibodies. (**D**) Number of PRMT5 peaks in BLM and control group HepG2 cells; (**E**) Different PRMT5 peaks in BLM group versus control group HepG2 cells. (**F**) Cells were treated with 5 μM, 10 μM, and 20 μM PRMT5 inhibitor GSK591 for 48 h to inhibit histone arginine methylation, H3K27ac modification level detected via immunoblot assay was decreased in a dose-dependent manner. (**G**) Histone H3R26me2s and H3K27ac levels were decreased in the *PRMT5* knockdown HepG2 cells. (**H**) Relative changes in H3R26me2s, H3K27ac, and H3K27me3 in *PRMT5* knockdown HepG2 cells (**G**). (**I**) Validation of H3R26me2s and H3K27ac levels change in the PRMT5 overexpressed cells through immunoblot. (**J**) Relative changes in H3R26me2s, H3K27ac, and H3K27me3 in PRMT5 overexpressed HepG2 cells (**I**). Data information: Data were analyzed with a two-tailed unpaired Student’s *t*-test. *: *p* < 0.05, **: *p* < 0.01, ***: *p* < 0.001.

**Figure 6 ijms-25-07562-f006:**
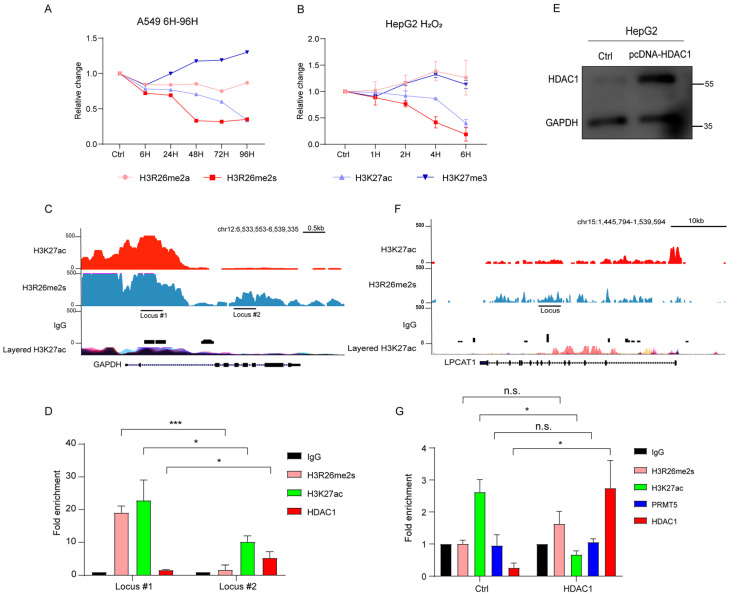
H3K27ac deacetylation mediated by HDAC1. (**A**) Normalized H3K27ac, H3K27me3, H3R26me2a, and H3R26me2s levels in bleomycin-treated A549 cells; A549 cells were treated with a single BLM concentration in a time-course manner, immune blot results are provided in Appendix A. (**B**) Normalized H3K27ac, H3K27me3, H3R26me2a, and H3R26me2s levels in H_2_O_2_-treated HepG2 cells; HepG2 cells were treated with a single H_2_O_2_ concentration in a time-course manner, immune blot results are provided in Figure 2C. (**C**) Respective of the H3R26me2s and H3K27ac modifications landscape of the *GAPDH* genomic locus. (**D**) H3R26me2s, H3K27ac, and HDAC1 enrichment in *GAPDH* genomic locus. (**E**) Validation of HDAC1 overexpression in HepG2 cells. (**F**) Respective of the H3R26me2s and H3K27ac modifications landscape of the *LPCAT1* genomic locus. (**G**) H3R26me2s, H3K27ac, PRMT5, and HDAC1 enrichment in *LPCAT1* genomic locus with control or HDAC1 overexpression cells. Data information: Data are shown as the mean ± SEM of three biological replicates and were analyzed with two-tailed unpaired Student’s *t*-test. *: *p* < 0.05, ***: *p* < 0.001, n.s.: no significance. Layered H3K27ac indicates H3K27ac ChIP-seq peaks, each color represents one cell line, and data were downloaded from the UCSC Genome Browser.

**Figure 7 ijms-25-07562-f007:**
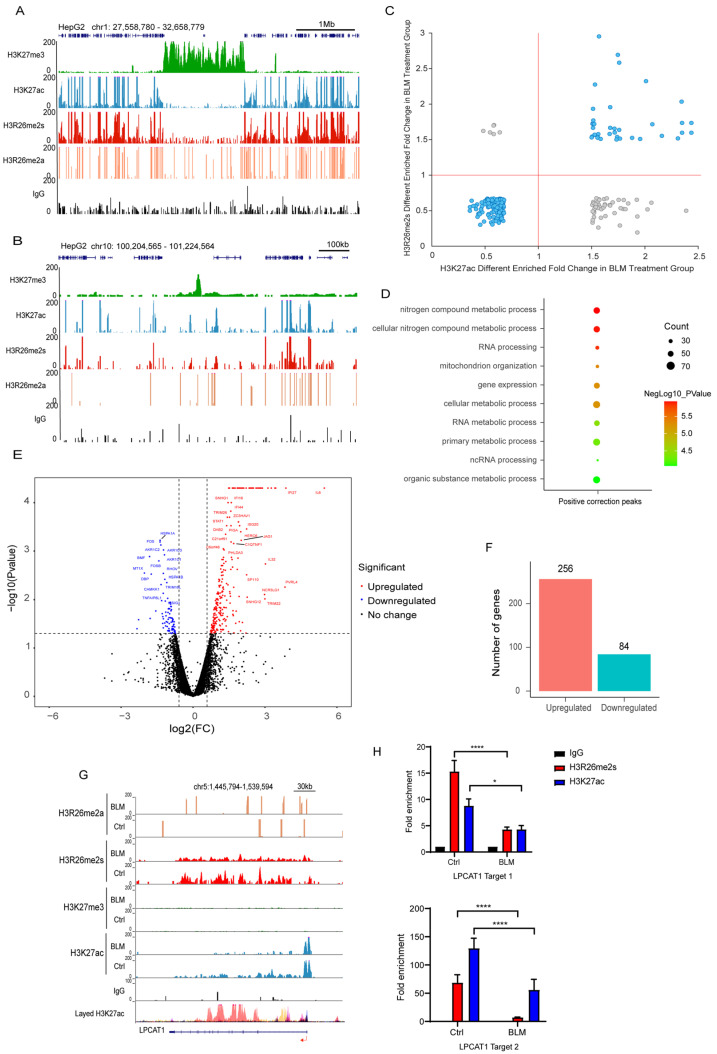
H3R26me2s shares a similar pattern with H3K27ac. (**A**) Respective of the H3R26me2a, H3R26me2s, H3K27ac, and H3K27me3 modifications landscape across a 1 Mb segment of the human genome; H3R26me2s shares a similar distribution pattern with H3K27ac and antagonizes H3K27me3. (**B**) Respective of the H3R26me2a, H3R26me2s, H3K27ac, and H3K27me3 modifications landscape across a 100 Kb segment of the human genome; H3R26me2s shares a similar distribution pattern with H3K27ac and antagonizes H3K27me3. (**C**) Overlapped H3R26me2s and H3K27ac up or down peaks; the cyan dots represent that H3R26me2s and H3K27ac have the same change pattern, while the grey dots represent that they have an opposite change pattern. (**D**) Gene Oncology of H3R26me2s and H3K27ac overlapped positive correction peaks shown in Figure 4F associated genes. (**E**) Volcano Plot of 256 downregulated and 84 upregulated genes in bleomycin-treated cells compared with control cells. (**F**) Number of genes up and down in bleomycin-treated HepG2 cells; gene fold changes above two were in the account. (**G**) Respective of the control and bleomycin-treated cells histone H3R26me2a, H3R26me2s, H3K27me3, and H3K27ac modifications in the downregulated gene *LPCAT1* locus. (**H**) H3R26me2s and H3K27ac modifications were decreased in bleomycin-treated cells at the *LPCAT1* locus quantitated using CUT&Tag-qPCR. Data information: Data are shown as the mean ± SEM of three biological replicates and were analyzed with two-tailed unpaired Student’s *t*-test. *: *p* < 0.05, ****: *p* < 0.0001.

**Figure 8 ijms-25-07562-f008:**
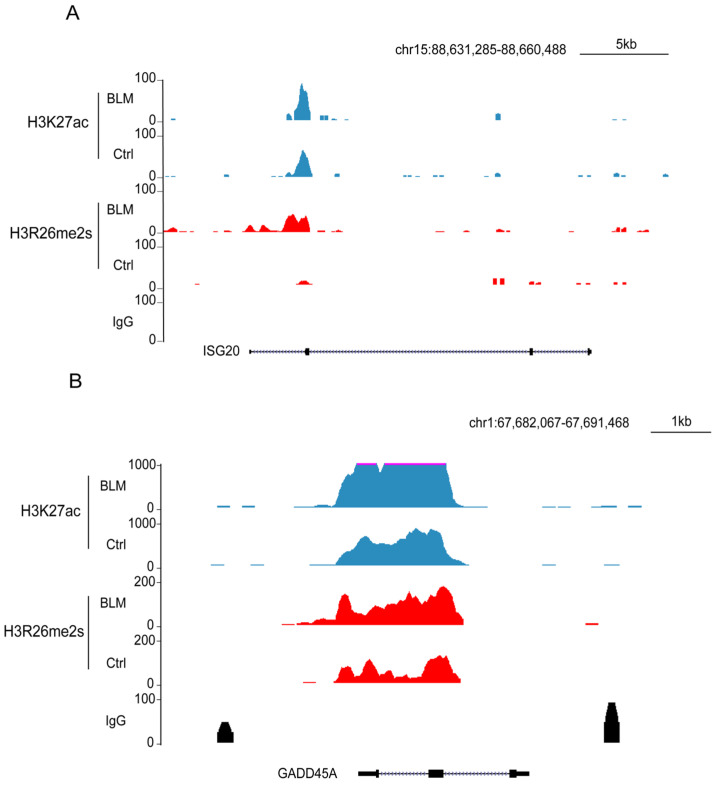
Feature and function of H3R26me2s and H3K27ac overlapped peaks. (**A**) H3R26me2s and H3K27ac peaks were increased in the upregulated gene *ISG15* locus. (**B**) H3R26me2s and H3K27ac peaks were increased in the upregulated gene *GADD45A* locus.

**Figure 9 ijms-25-07562-f009:**
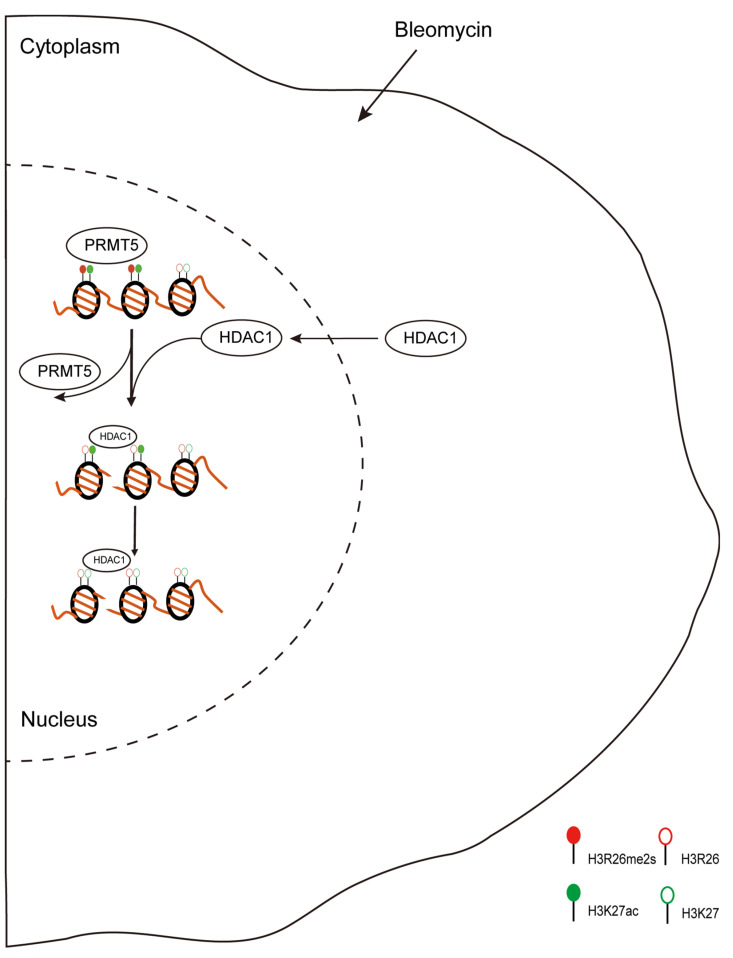
Proposed scheme for DNA damage-induced histone modifications change. During DNA damage, histone modifications change to regulate gene expression or DNA repair. In this study, we found DNA damage-induced histone arginine demethylation and lysine deacetylation. Moreover, the lower H3R26me2s region then recruited HDAC1 to catalyze H3K27 deacetylation. H3R26me2s and H3K27ac have similar occupancy across the genome, indicating the crosstalk between H3R26me2s and H3K27ac in regulating gene expression.

## Data Availability

RNA-seq and CUT&Tag-seq data have been deposited at GEO and are publicly available. GEO number: GSE239630.

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
