# Peer review of "Mechanism of Histone Arginine Methylation Dynamic Change in Cellular Stress"

_ijms, 2024, doi:10.3390/ijms25147562_

Round 1

Reviewer 1 Report

Comments and Suggestions for Authors

The manuscript provides a thorough and insightful analysis of the dynamic changes in histone arginine methylation, specifically focusing on H3R26me2s, in response to cellular stress. The relationship between HDAC1 and H3R26me2s is particularly novel and adds significant value to the field of epigenetics and stress response mechanisms. The manuscript is well-organized and clearly presents the findings in a logical manner, making it accessible to readers.

Here are few additions which can be incorporated in the paper:

Expand the discussion section and address the address the broader implications of your findings. How might this interaction influence other known pathways in stress response? Are there potential clinical or therapeutic applications?

Reviewer 2 Report

Comments and Suggestions for Authors

The paper “Mechanism of Histone Arginine Methylation Dynamic Change in Cellular Stress” aimed to investigate the changes in histone arginine methylation in response to DNA damage. Here are some questions that need to be further improved or explained.

Comments:

Q1. What are the advantages of choosing this drug (bleomycin BLM)?.

Q2. Has the mechanism of BLM-induced DNA damage in cells been reported in detail?

Q3. There is no difference between the original images and the WB images presented in the paper, and the original images are not full gels.

Q4. The grayscale of many WB bands has not been well analyzed. No difference significance analysis could be performed with values missing from parallel trials.

Q5. In Figure 2, the magnification times of (D) and (E) are different? Besides, what does the distributions of histone acetylation and methylation tell us? Why does DNA damage occur, but karyopyknosis and numerous apoptotic bodies are not observed?

Q6. Supplementary conclusions are suggested to further summarize the results of the full text. Why is a large amount of data put in the supplementary file, is it meaningless to support the conclusion of the article?

On the whole, this paper is rich in data and heavy in workload. If the above problems can be properly solved, the paper could be suggested for publication.

Round 2

Reviewer 2 Report

Comments and Suggestions for Authors

The repetition rate is still a little high.

Author Response

Dear Reviewer,

We thank you for the critical comments and helpful suggestions,wWe have studied comment carefully and have made corrections which we hope meet with approval.

Revision notes, point-to-point, are given as follows

1. The repetition rate is still a little high.

Reply: Thanks for your available suggestion, we apologize for our mistake in repeated writing, we have deleted the related words and replaced them with new sentences in the new manuscript.